# MathHay: An Automated Benchmark for Long-Context Mathematical Reasoning in LLMs

## Abstract

Recent large language models (LLMs) have demonstrated versatile capabilities in long-context scenarios. Although some recent benchmarks have been developed to evaluate the long-context capabilities of LLMs, there is a lack of benchmarks evaluating the mathematical reasoning abilities of LLMs over long contexts, which is crucial for LLMs' application in real-world scenarios. In this paper, we introduce MathHay, an automated benchmark designed to assess the **long-context mathematical reasoning capabilities** of LLMs. Unlike previous benchmarks like Needle in a Haystack, which focus primarily on information retrieval within long texts, MathHay demands models with both information-seeking and complex mathematical reasoning abilities. We conduct extensive experiments on MathHay to assess the long-context mathematical reasoning abilities of eight top-performing LLMs. Even the best-performing model, Gemini-1.5-Pro-002, still struggles with mathematical reasoning over long contexts, achieving only 51.26% accuracy at 128K tokens. This highlights the significant room for improvement on the MathHay benchmark.

## 1 Introduction

Long-context tasks arise in various applications, including summarization (Huang et al., 2021), multi-document question answering (Yang et al., 2018), prompt compression (Jiang et al., 2023a;b), and repository-level code generation (Bogomolov et al., 2024). Recent large language models (LLMs) such as GPT-4 (OpenAI, 2023), Claude (Claude, 2023), and Gemini (Reid et al., 2024) have shown versatile capabilities across various long-context scenarios. They are designed to support long context modeling, being able to process up to 128k or even 2M tokens (Reid et al., 2024).

Some recent benchmarks have been developed to evaluate the long-context capabilities of LLMs. *LongBench* (Bai et al., 2023) is a benchmark that covers 6 tasks, with an average length of about 7,000 words (English version). To evaluate the ability of LLMs to handle longer contexts, *Needle in a Haystack* (Kamradt, 2023) is increasingly popular. This test requires models to locate a small, specific piece of information within varying long context windows. However, recent advanced LLMs can easily achieve near-perfect performance on Needle in a Haystack (Dubey et al., 2024). To refine the evaluation of long-context ability LLMs, several variants of the Needle in a Haystack task have been introduced. For example, Laban et al. (2024) presents *Summary of a Haystack*, a summarization-based test that evaluates reasoning over long contexts and the ability to grasp content importance. *NeedleBench* (Li et al., 2024) positions critical data points at varying depths within texts, testing retrieval and reasoning abilities in contexts ranging from 4k to 1000k tokens. In addition, the *BABILong benchmark* (Kuratov et al., 2024) is designed to test models' reasoning across facts dispersed throughout extremely long documents, encompassing 20 tasks such as fact chaining, induction, deduction, counting, and managing lists/sets.

While these benchmarks bring complexity and diversity to evaluate the capabilities of the latest LLMs in long-context scenarios, there is still a lack of appropriate benchmarks for evaluating their long-context abilities in mathematical reasoning, which often arise in real-world situations. For example, some example scenarios where such long-context mathematic reasoning can be helpful for users i) if there is a set of news about Nvidia's Q2 in 2024, then the user might want to know how much revenue increased compared to the previous quarter, or the earnings per share for the quarter, and whether they exceeded analysts' expectations ii) the user wants to compare Microsoft's and Amazon's

| Benchmark | Multi-Doc Tasks | Multi-Step Reasoning | Avoidance of Contamination | Irrelevant Documents | Realistic Documents | Automated Construction | Mathematical Reasoning |
|---|---|---|---|---|---|---|---|
| ZeroSCROLLS (Shaham et al., 2023) | ✓ | ✓ | ✗ | ✓ | ✓ | ✗ | ✗ |
| L-Eval (Math) (An et al., 2023) | ✓ | ✗ | ✗ | ✗ | ✗ | ✗ | ✓ |
| LongBench (Bai et al., 2023) | ✓ | ✗ | ✗ | ✓ | ✓ | ✗ | ✗ |
| BAMBOO (Dong et al., 2023) | ✗ | ✗ | ✓ | ✓ | ✓ | ✗ | ✗ |
| InfiniteBench (Math) (Zhang et al., 2024) | ✓ | ✓ | ✗ | ✓ | ✗ | ✗ | ✓ |
| Loong (Wang et al., 2024) | ✓ | ✓ | ✗ | ✓ | ✓ | ✗ | ✗ |
| NIAH (Kamradt, 2023) | ✗ | ✗ | ✗ | ✓ | ✓ | ✓ | ✗ |
| RULER (Hsieh et al., 2024) | ✓ | ✓ | ✗ | ✓ | ✓ | ✓ | ✗ |
| FlenQA (Levy et al., 2024) | ✓ | ✓ | ✗ | ✓ | ✓ | ✓ | ✗ |
| SummHay (Laban et al., 2024) | ✓ | ✗ | ✗ | ✓ | ✓ | ✗ | ✗ |
| BABILong (Kuratov et al., 2024) | ✓ | ✓ | ✗ | ✓ | ✓ | ✓ | ✗ |
| NeedleBench (Li et al., 2024) | ✓ | ✓ | ✗ | ✓ | ✓ | ✓ | ✗ |
| MATHHAY (Ours) | ✓ | ✓ | ✓ | ✓ | ✓ | ✓ | ✓ |

Table 1: Comparative analysis of MATHHAY and existing long-context benchmarks.

cloud income and expenditure in Q2 of 2024 to help determine if they should invest in Microsoft or Amazon stocks. iii) Knowing the population growth rates for previous year and this year in a certain country can help the user decide whether to invest in real estate there in the future. For these real-world queries, there is need for the ability to gather extensive materials from different sources, identify the precise relevant information within it and perform some mathematical reasoning in order to derive the correct answer. This inspires us to create a new mathematical reasoning benchmark to evaluate LLMs' long-context capabilities in more real-world scenarios.

In this paper, we introduce MATHHAY, an automated benchmark designed to evaluate long-context mathematical reasoning in LLMs. The benchmark is built through four key stages: document collection, question generation, quality control, and haystack construction. First, we gather documents featuring real-world mathematical reasoning scenarios within a certain time period to support to form MATHHAY. Next, we generate four types of test tasks, varying in difficulty: (1) Single-Step, Single-Document (SSSD), (2) Multi-Step, Single-Document (MSSD), (3) Single-Step, Multi-Document (SSMD), and (4) Multi-Step, Multi-Document (MSMD). SSSD is the simplest, requiring a single relevant document and one computational step, while MSMD is the most complex, requiring multiple documents and computational steps. After question generation, we apply quality control by comparing solutions generated through different strategies to ensure high-quality data. Finally, we construct the haystack for MATHHAY by inserting relevant documents into noisy text using certain placement strategies. Our main contributions are summarized as follows:

- We introduce an automated method to create high-quality long-context mathematical reasoning benchmarks tailored for real-world scenarios within a specified time period.
- We present the MATHHAY benchmark, which includes questions of varying difficulty levels to assess LLMs' reasoning abilities across different input lengths (32K, 64K, 128K).
- We conduct extensive experiments on MATHHAY to assess the long-context reasoning abilities of eight top-performing LLMs. Our results show that current LLMs struggle to handle mathematical reasoning tasks over long contexts, highlighting significant room for improvement on the MATHHAY benchmark.

## 2 RELATED WORK

### 2.1 LONG-CONTEXT BENCHMARKS

Long-context modeling is rapidly growing, with several benchmarks developed to evaluate this capability by building on or revising existing tasks and datasets. ZeroSCROLLS (Shaham et al., 2023) facilitates systematic comparisons of LLMs on tasks requiring information from long texts. LongBench (Bai et al., 2023) introduces a multitask bilingual benchmark for long-context understanding, spanning 21 tasks. Loong (Wang et al., 2024) highlights a key limitation of current benchmarks that artificially extend input lengths with irrelevant noise. Loong aims to reflect real-world scenarios through extended multi-document question answering. BAMBOO (Dong et al., 2023) addresses data contamination in long-context settings by incorporating more recent documents into the benchmark. L-Eval (An et al., 2023) offers a comprehensive suite of tasks for long-context models.

InfiniteBench (Zhang et al., 2024) is the first benchmark featuring data lengths exceeding 100K tokens. L-Eval and InfiniteBench include mathematical reasoning tasks, but MATHHAY stands out by introducing irrelevant documents, making reasoning more challenging. Needle-in-a-Haystack (NIAH) (Kamradt, 2023) evaluates LLM recall by embedding a fact within long contexts but focuses on shallow understanding. RULER (Hsieh et al., 2024) builds on NIAH with more complex tasks involving multi-hop reasoning. SummHay (Laban et al., 2024) focuses on summarizing large document sets, while BABILong (Kuratov et al., 2024) tests reasoning across dispersed facts in long documents. NeedleBench (Li et al., 2024) provides a customizable framework for bilingual long-context evaluations. In addtion, DocFinQA (Reddy et al., 2024) is developed to assess financial reasoning in LLMs and DOCMATH-EVAL (Zhao et al., 2023) is manually annotated by experts to evaluate the mathematical reasoning abilities of LLMs within a context length of 35K. Compared to these benchmarks, MATHHAY is designed to automatically evaluate LLMs' mathematical reasoning in longer, more diverse, and real-world contexts.

## 2.2 MATHEMATICAL REASONING BENCHMARKS

Assessing mathematical reasoning abilities is crucial for advancing large language models. Early work in this area includes MathQA (Amini et al., 2019), which introduces a "large-scale" dataset of math word problems densely annotated with operation programs, curated from the AQuA (Ling et al., 2017) dataset. Later, GSM8K (Cobbe et al., 2021) and MATH (Hendrycks et al., 2021) provide high-quality datasets of linguistically diverse grade school problems and challenging competition-level problems, respectively. These datasets, known for their difficulty, are widely used to evaluate mathematical reasoning capabilities of large language models. More recent efforts, such as LILA (Mishra et al., 2022), introduce a unified benchmark of 23 mathematical reasoning tasks across multiple dimensions, further expanding the evaluation of AI systems in mathematics. GHOSTS (Frieder et al., 2024) shifts the focus towards graduate-level math, addressing professional use cases for models like GPT-4 in assisting mathematicians. Our benchmark, MATHHAY, extends the exploration to long-context scenarios, focusing on multi-step mathematical reasoning, making it a unique contribution to benchmarking the mathematical reasoning abilities of large language models over long contexts.

## 3 BENCHMARK CONSTRUCTION

In this section, we go through the steps taken to automatically construct the MATHHAY benchmark and ensure the quality of the constructed benchmark. Figure 1 illustrates the automated process, which consists of four main stages: document collection, question generation, quality control, and haystack construction. We provide a detailed explanation of each step in this section.

### 3.1 DOCUMENT COLLECTION

The document collection stage involves gathering texts from sources that potentially include mathematical reasoning in real-world scenarios. These documents should contain sufficient numerical values to construct data examples for the MATHHAY benchmark.

**Topic Generation.** We aim for MATHHAY to cover diverse topics, including Financial Market Analysis, Sports Performance Metrics, and Climate Change Impact Assessment, where queries frequently require mathematical reasoning. To facilitate this, we designed a prompt to guide the LLM in generating responses on these topics. Refer to the corresponding prompt in Appendix A.1.1.

**Relevant Document Collection.** After obtaining key topics related to mathematical reasoning, we prompt the LLM to generate subtopics along with corresponding queries. Each subtopic is paired with several specific queries. For example, under the "Nvidia's stock price" subtopic, a potential query could be, "Compare Nvidia's end-of-month stock prices for April 2024 and May 2024". To ensure the queries are time-sensitive, we incorporate a time period constraint in the prompt, guiding the LLM to generate queries within a specific time range. This keeps the MATHHAY benchmark up-to-date and may help mitigate data leakage (test data from a benchmark might be included in the training set of newer models (White et al., 2024)), enabling a fairer evaluation of different LLMs' abilities. For this benchmark, we set the time period from January to August 2024. Refer to the corresponding prompt in Appendix A.1.2.

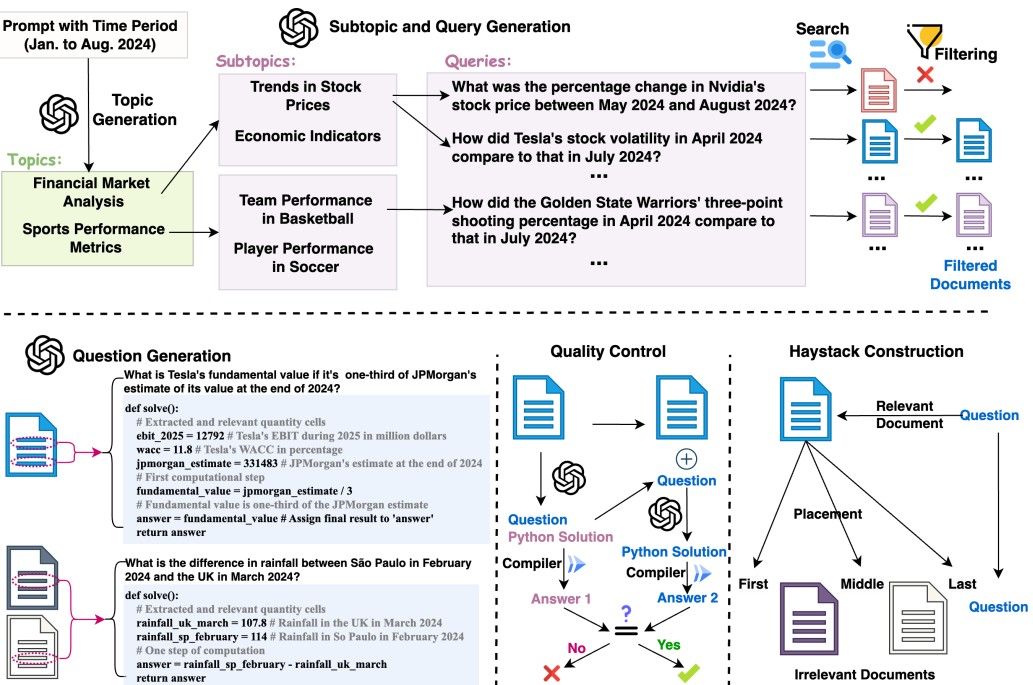

Figure 1: Overview of the framework for the automatic construction of the MATHHAY Benchmark. The upper section illustrates the document collection process, while the lower section outlines the stages of question generation, quality control, and haystack construction.

The generated queries are used to retrieve relevant documents from online sources. For each query, we employ Tavily Search[1] to gather up-to-date and relevant information. From the search results, we select the top-ranked document as the most relevant for each query.

**Document Filtering.** After gathering the initial set of documents from search engine, we implement a filtering process to retain sufficient numerical values and informative texts for constructing high-quality mathematical reasoning problems. First, each document has to contain more than a specific number of distinct numerical values (excluding dates) to ensure sufficient complexity for generating diverse, multi-step reasoning problems. Documents with fewer numbers might be inadequate for testing LLMs' numerical reasoning abilities. Second, we prioritized documents with rich context, including ample sentences, sufficient words, and diverse named entities such as people, places, and organizations. This ensured that the later generated questions could be grounded in real-world scenarios. Through this process, we narrowed the collected documents to a refined set of high-quality documents rich in numerical values and contextual depth, enabling the generation of more realistic and challenging reasoning problems.

## 3.2 QUESTION GENERATION

To construct a comprehensive benchmark for evaluating models' capabilities in long-context mathematical reasoning, we designed a series of test tasks that vary in difficulty. The tasks can be divided into four distinct categories: (1) Single-Step, Single-Document Mathematical Reasoning Task, (2) Multi-Step, Single-Document Mathematical Reasoning Task, (3) Single-Step, Multi-Document Mathematical Reasoning Task, and (4) Multi-Step, Multi-Document Mathematical Reasoning Task.

**Single-Step, Single-Document Mathematical Reasoning Task (SSSD).** Questions in this task require a single computational step $(+, -, \times, \div)$ to reach the solution, based on information contained within a single document. This task assesses the model's ability to extract relevant numerical information from a single document within a document haystack and perform mathematical reasoning

---

[1]Tavily Search is a search engine optimized for LLMs and RAG, designed for efficient, fast, and persistent results. More information is available at https://tavily.com/

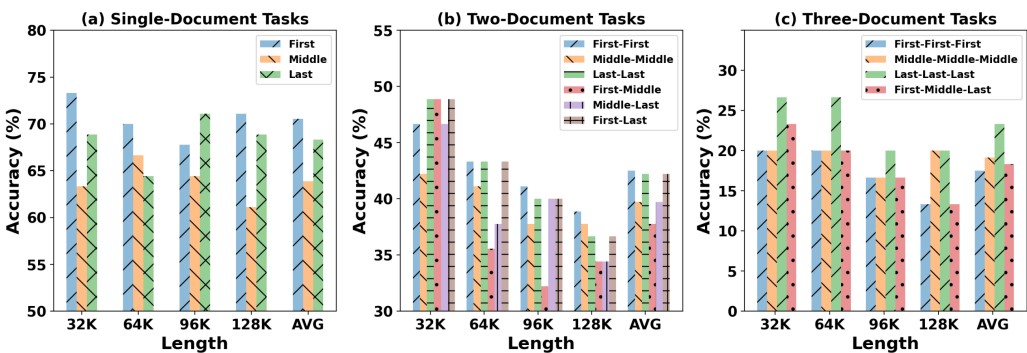

Figure 2: Accuracy of GPT-4o-mini on (a) single-document; (b) two-document; (c) three-document mathematical reasoning tasks from a subset of the MATHHAY Benchmark, with varying relevant document placements and input lengths.

to arrive at a correct answer. The LLM is prompted to generate the question and Python solution (*i.e.*, the solution process represented as a Python program). Refer to the corresponding prompt in Appendix A.1.3.

**Multi-Step, Single-Document Mathematical Reasoning Task (MSSD).** This task involves questions requiring multiple computational steps to reach a solution, based on information within a single document. Unlike the SSSD, the MSSD challenges the model to identify multiple snippets containing numerical data and then correctly sequence them into intermediate reasoning steps. An LLM is used to generate the question and a one-step solution process as a Python program.

**Single-Step, Multi-Document Mathematical Reasoning Task (SSMD).** In this category, the task requires the model to solve a problem that involves information spread across multiple documents. Although the solution involves only a single computational step, the complexity lies in the need to correctly identify and extract relevant numerical values from different documents in the haystack. The LLM is prompted to generate the question and the one-step Python solution.

**Multi-Step, Multi-Document Mathematical Reasoning Task (MSMD).** This category represents the most complex task, challenging models to perform multi-step reasoning and extract information from multiple documents. It requires the model to sequentially process and combine numerical values from several sources, while maintaining clear mathematical reasoning and accuracy throughout the calculations. The LLM is prompted to generate the question and Python solution.

### 3.3 QUALITY CONTROL

Given the range of tasks in the MATHHAY benchmark, which span from single-step to multi-step reasoning across one or more documents, it's crucial that the solution process produces the correct final answer. To ensure the quality of the generated data examples, we implement a quality control process that focuses on consistency across different solutions for each question.

The quality control process begins by executing the Python solution generated by the LLM from the previous question-generation stage using a Python interpreter to get the first answer. Next, we re-feed the question and relevant documents into the LLM, prompting it to generate another Python solution. This second solution is also executed to produce a new answer. We then compare the two answers: if they match, the example is considered as high quality and suitable to be included in the benchmark. If the answers differ, the example is filtered out for being inconsistent. Refer to corresponding prompts in Appendix A.1.4.

### 3.4 HAYSTACK CONSTRUCTION

To accurately assess models' ability to handle long-context mathematical reasoning, we construct document "haystacks" of varying sizes, simulating real-world scenarios where relevant information is

buried within large volumes of irrelevant data. This setup challenges the models to filter out noise and identify the necessary details needed to solve the problem. We vary the sizes of the haystacks, with token lengths ranging from 32K to 128K tokens. Each haystack contains a mixture of documents: a small number of question-relevant documents (one relevant document for one-document reasoning tasks) and a larger pool of irrelevant ones, which are actually relevant to other unrelated queries. These unrelated queries mainly come from different topics. This design ensures that only a few documents in each haystack are helpful for answering the target question, making the task progressively more difficult as the haystack size increases.

We implement different placement strategies when inserting relevant documents into irrelevant documents. For single-document reasoning tasks, we experiment with three strategies: (1) **First**: The relevant document is placed at the beginning of the irrelevant documents, which are furthest from the target question; (2) **Middle**: The relevant document is inserted in the middle of the irrelevant documents; (3) **Last**: The relevant document is appended to the end of the irrelevant documents.

For two-document reasoning tasks, where two relevant documents are needed to solve the problem, we expand the placement strategies to combinations of positions: (1) **First-First**: Both relevant documents are placed at the beginning; (2) **Middle-Middle**: Both relevant documents are placed in the middle; (3) **Last-Last**: Both relevant documents are placed at the end; (4) **First-Middle**: One relevant document is placed at the beginning, and the second in the middle; (5) **Middle-Last**: One relevant document is placed in the middle, and the second at the end; (6) **First-Last**: One relevant document is placed at the beginning, and the other at the end.

For three-document reasoning tasks, the complexity of document placement further increases. We introduce the following four combinations: (1) **First-First-First**: All three relevant documents are placed at the beginning; (2) **Middle-Middle-Middle**: All three relevant documents are placed in the middle; (3) **Last-Last-Last**: All three relevant documents are placed at the end; (4) **First-Middle-Last**: The three relevant documents are distributed evenly, one at the beginning, one in the middle, and one at the end.

Figure 2 shows GPT-4o-mini's accuracy on single, two, and three-document mathematical reasoning tasks, varying by document placement and input length. We can observe that the middle placement is most challenging for single-document tasks, first-middle for two-document tasks, and first-first-first for three-document tasks. Based on these results, we select these placements for each task type in constructing the MATHHAY benchmark.

## 3.5 STATISTICS OF MATHHAY BENCHMARK

Table 2 presents the main statistics of MATHHAY, and Figure 3 shows the topic and task distribution of MATHHAY. The dataset includes 673 questions across 10 topics and 40 subtopics, with 233 single-step, 168 two-step, and 198 three-step reasoning tasks. On average, each question contains 33.31 words and is linked to 1.53 relevant documents, with the average document length being 4190.53 tokens. The average number of reasoning steps per question is 2.00.

The dataset is divided into verified and unverified questions. Verified data refers to questions that have been reviewed by authors to ensure the correctness of the reasoning steps. Incorrect data examples are removed, while correct ones are retained. Of the 126 verified questions, 52 are single-step, and their average length is 35.25 words. These questions are linked to 1.58 relevant documents, averaging 4139.37 tokens in length, with 1.85 reasoning steps per question. The unverified portion, with 547 questions, has an average length of 32.87 words. These questions are linked to 1.52 relevant documents, averaging 4202.31 tokens, and require an average of 2.03 reasoning steps.

# 4 EXPERIMENT

## 4.1 EXPERIMENTAL SETUP

**Models.** In this work, we evaluate several cutting-edge long-context LLMs using the proposed MATHHAY Benchmark. Our evaluation includes both closed-source and open-source models, tested across varying token lengths: 32K, 64K, and 128K. For the closed-source models, we assess the

| Statistic | Number |
|---|---|
| Time period | Jan. to Aug. 2024 |
| Topics | 10 |
| Subtopics | 40 |
| Questions | 673 |
| Single-step questions | 233 |
| Two-step questions | 168 |
| Three-step questions | 198 |
| Avg. question length | 33.31 |
| Avg. relevant documents | 1.53 |
| Avg. relevant document length | 4190.53 |
| Avg. reasoning steps | 2.00 |
| Verified | |
| - Questions | 126 |
| - Single-step questions | 52 |
| - Two-step questions | 0 |
| - Three-step questions | 0 |
| - Avg. question length | 35.25 |
| - Avg. relevant documents | 1.58 |
| - Avg. relevant document length | 4139.37 |
| - Avg. reasoning steps | 1.85 |
| Unverified | |
| - Questions | 547 |
| - Single-step questions | 181 |
| - Two-step questions | 168 |
| - Three-step questions | 198 |
| - Avg. question length | 32.87 |
| - Avg. relevant documents | 1.52 |
| - Avg. relevant document length | 4202.31 |
| - Avg. reasoning steps | 2.03 |

Table 2: Key statistics of MATHHAY.

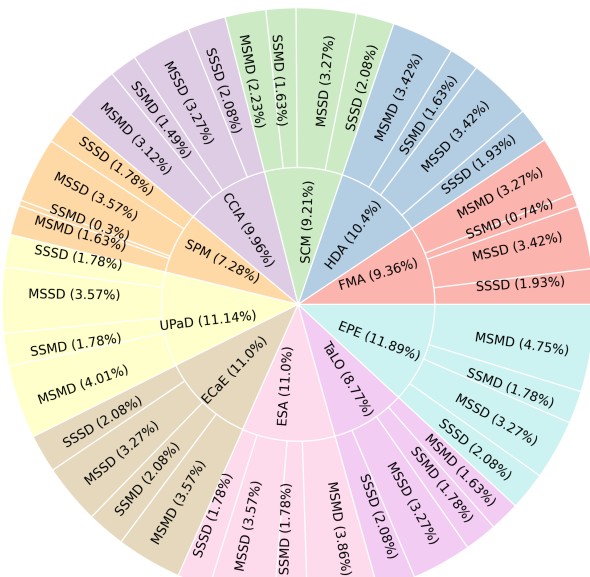

Figure 3: Topic and task distribution. FMA: Financial Market Analysis, HCA: Healthcare Cost Analysis, UP: Urban Planning, EIA: Environmental Impact Assessment, SCM: Supply Chain Management, SA: Sports Analytics, ECA: Energy Consumption Analysis, REMT: Real Estate Market Trends, EF: Education Funding, AE: Agricultural Economics.

performance of several models from the GPT series[2] (OpenAI, 2023; 2024a;b), including GPT-4o (128K), GPT-4o-Mini (128K), o1-preview, and o1-mini, Claude-3.5-Sonnet[3] (Anthropic, 2024), and Gemini-1.5-Pro-002[4] (Reid et al., 2024). On the open-source side, we evaluate Qwen-2.5-7B-Instruct (128K) (Team, 2024) and LLaMA-3.1-8B-Instruct (128K) (Dubey et al., 2024), two recent advanced models in the open research community.

**Evaluation.** In mathematical reasoning tasks, LLMs often generate long explanations instead of directly providing numerical values as final answers. This poses challenges for traditional evaluation methods, such as rule-based or template-based exact match, which struggle to accurately assess the output. To address this, some benchmarks have adopted the practice of using LLMs as judges (Lu et al., 2023). Building on this, we combine rule-based exact matches with LLM judgment to assess the correctness of generated answers. We chose GPT-4o as our evaluation judge due to its advanced reasoning and assessment capabilities (Dubois et al., 2024). If an exact match is achieved, the predicted answer is considered correct, and a score of 1 is assigned. In cases where the exact match fails, we rely on the LLM judge. If the LLM deems the answer correct, we also consider the predicted answer correct. Conversely, if the LLM judges the answer to be incorrect, it is marked as wrong. A preliminary study of 100 examples demonstrates that GPT-4o, when used as a judge, correlates almost perfectly with human evaluations in our benchmark. Detailed instructions for this evaluation process are provided in Appendix A.2.

**Implement Details.** We set the temperature to zero for all models to ensure deterministic predictions. For closed-source models, we use the provided API for testing. For open-source models, we use vLLM[5] to build service to provide API for testing using NVIDIA A100 (40GB) GPUs.

---

[2]https://openai.com/api/

[3]https://claude3.pro/claude-3-5-sonnet-api/

[4]https://aistudio.google.com/app/apikey

[5]https://github.com/vllm-project/vllm

| Model | Claimed Length | SSSD | | MSSD | | SSMD | | MSMD | | Overall | | |
|---|---|---|---|---|---|---|---|---|---|---|---|---|
| | | Verified | Unverified | Verified | Unverified | Verified | Unverified | Verified | Unverified | Verified | Unverified | Full |
| *32K* | | | | | | | | | | | | |
| LLaMA-3.1-8B-Instruct | 128K | 40.62 | 44.00 | 28.57 | 27.5 | 35.00 | 20.99 | 15.22 | 17.47 | 27.78 | 26.51 | 26.75 |
| Qwen-2.5-7B-Instruct | 128K | 46.88 | 52.00 | 32.14 | 27.00 | 50.00 | 34.57 | 6.52 | 21.08 | 29.37 | 30.89 | 30.61 |
| GPT-4o-mini | 128K | 71.88 | 68.00 | 42.86 | 42.50 | 50.00 | 50.62 | 26.09 | 35.54 | 45.24 | 46.25 | 46.06 |
| GPT-4o | 128K | 71.88 | 73.00 | 53.57 | 53.50 | 60.00 | 55.56 | 34.78 | 45.18 | 52.38 | **54.85** | 54.38 |
| o1-mini | 128K | 56.25 | 68.00 | 50.00 | 51.00 | 60.00 | 48.15 | 34.78 | 35.54 | 47.62 | 48.81 | 48.59 |
| o1-preview | 128K | 62.50 | 69.00 | 50.00 | 51.00 | 65.00 | 46.91 | 30.43 | 34.34 | 48.41 | 48.63 | 48.59 |
| Claude-3.5-Sonnet | 200K | 68.75 | 77.00 | 46.43 | 53.00 | 65.00 | 51.85 | 32.61 | 39.16 | 50.00 | 53.02 | 52.45 |
| Gemini-1.5-Pro-002 | 2M | 68.75 | 75.00 | 57.14 | 52.00 | 70.00 | 44.44 | 32.61 | 37.95 | **53.17** | 50.82 | 51.26 |
| *64K* | | | | | | | | | | | | |
| LLaMA-3.1-8B-Instruct | 128K | 53.12 | 58.00 | 39.29 | 30.00 | 35.00 | 24.69 | 10.87 | 19.28 | 31.75 | 31.08 | 31.20 |
| Qwen-2.5-7B-Instruct | 128K | 28.12 | 45.00 | 21.43 | 24.50 | 30.00 | 28.40 | 6.52 | 21.69 | 19.05 | 27.97 | 26.30 |
| GPT-4o-mini | 128K | 59.38 | 63.00 | 39.29 | 38.00 | 60.00 | 45.68 | 21.74 | 31.33 | 41.27 | 41.68 | 41.61 |
| GPT-4o | 128K | 65.62 | 69.00 | 53.57 | 48.50 | 65.00 | 48.15 | 32.61 | 40.96 | 50.79 | 49.91 | 50.07 |
| o1-mini | 128K | 56.25 | 60.00 | 60.71 | 47.00 | 65.00 | 50.62 | 26.09 | 33.13 | 47.62 | 45.70 | 46.06 |
| o1-preview | 128K | 59.38 | 71.00 | 42.86 | 52.00 | 65.00 | 46.91 | 28.26 | 36.75 | 45.24 | 50.09 | 49.18 |
| Claude-3.5-Sonnet | 200K | 53.12 | 67.00 | 53.57 | 50.50 | 60.00 | 48.15 | 34.78 | 36.75 | 47.62 | 49.00 | 48.74 |
| Gemini-1.5-Pro-002 | 2M | 68.75 | 73.00 | 57.14 | 53.50 | 70.00 | 50.62 | 32.61 | 38.55 | **53.17** | **52.10** | **52.30** |
| *128K* | | | | | | | | | | | | |
| LLaMA-3.1-8B-Instruct | 128K | 37.50 | 43.00 | 35.71 | 29.50 | 10.00 | 9.88 | 2.17 | 10.24 | 19.84 | 23.22 | 22.59 |
| Qwen-2.5-7B-Instruct | 128K | 15.62 | 26.00 | 14.29 | 16.50 | 20.00 | 14.81 | 10.87 | 7.23 | 14.29 | 15.17 | 15.01 |
| GPT-4o-mini | 128K | 56.25 | 65.00 | 32.14 | 39.50 | 35.00 | 39.51 | 21.74 | 30.12 | 34.92 | 41.32 | 40.12 |
| GPT-4o | 128K | 68.75 | 69.00 | 45.00 | 48.00 | 55.00 | 56.79 | 28.26 | 42.17 | 46.38 | 51.37 | 50.37 |
| o1-mini | 128K | 43.75 | 47.00 | 35.71 | 37.00 | 45.00 | 34.57 | 21.74 | 28.92 | 34.13 | 36.02 | 35.66 |
| o1-preview | 128K | 62.50 | 70.00 | 57.14 | 53.50 | 60.00 | 46.91 | 21.74 | 34.34 | 46.03 | 49.73 | 49.03 |
| Claude-3.5-Sonnet | 200K | 59.38 | 59.00 | 42.86 | 47.00 | 55.00 | 35.80 | 23.91 | 29.52 | 42.06 | 42.23 | 42.20 |
| Gemini-1.5-Pro-002 | 2M | 62.50 | 74.00 | 57.14 | 52.50 | 60.00 | 53.09 | 32.61 | 36.14 | **50.00** | **51.55** | **51.26** |

Table 3: Performance of Selected Models on MATHHAY (32K to 128K tokens). The model with the best performance is highlighted in bold.

## 4.2 RESULTS

We assess eight advanced LLMs on the MATHHAY benchmark, with the key results presented in Table 3. GPT-4o demonstrates the highest overall performance, achieving 54.38% at an input length of 32K. Gemini-1.5-pro-002 achieves the highest overall performance, reaching 52.30% at 64K and 51.26% at 128K. We can see that even one of the best-performing models, Gemini-1.5-Pro-002, still struggles with long contexts, achieving only 51.26% on the 128K input length, which is 48.74% below perfect accuracy. This performance gap highlights the significant room for improvement on the MATHHAY benchmark. In addition, to assess the quality of the automated MATHHAY benchmark (unverified), we computed the Spearman rank correlation between the human-verified and unverified MATHHAY. The resulting correlation coefficient of 0.9183 indicates a strong alignment in model rankings across unverified and verified test data, suggesting that the automated benchmark can reliably approximate the human-verified benchmark and be useful for evaluating models.

**Model Analysis:** We can observe that closed-source models perform relatively well compared to open-source models across all length settings. For instance, the best-performing open-source model, LLaMA-3.1-8B, achieves 22.59% accuracy at 128K. However, it still lags behind the worst-performing closed-source model, o1-mini, by 13.07%. These findings suggest that closed-source models excel in long-context mathematical reasoning compared to open-source counterparts.

**Task Analysis:** From the task perspective, models consistently perform better on simpler tasks. performance of models on single-step single-document tasks (SSSD) are much better than that of models on multi-step single-document tasks (MSSD), and models on single-step multi-document tasks (SSMD) show better performance than models multi-step multi-document tasks (MSMD). For example, GPT-4o reaches 71.88% accuracy on verified SSSD at 32K but drops to 53.57% on verified MSSD. Similarly, QWen-2.5-7B achieves 20.00% on verified SSMD at 128K but only 10.87% on MSMD in the same setting. These results suggest that tasks with multiple reasoning and computational steps are significantly more challenging, especially when large amounts of noisy text are involved. Furthermore, multi-step tasks across multiple documents (MSMD) are more difficult than those within a single document (MSSD), as evidenced by consistently lower performance on MSMD across all input lengths. This suggests that gathering information and reasoning across multiple documents is more challenging than doing so from a single document.

**Length Analysis:** While a few models demonstrate improved performance with longer input lengths (e.g., LLaMA-3.1-8B increases from 26.75% at 32K to 31.20% at 64K), most models show a decline

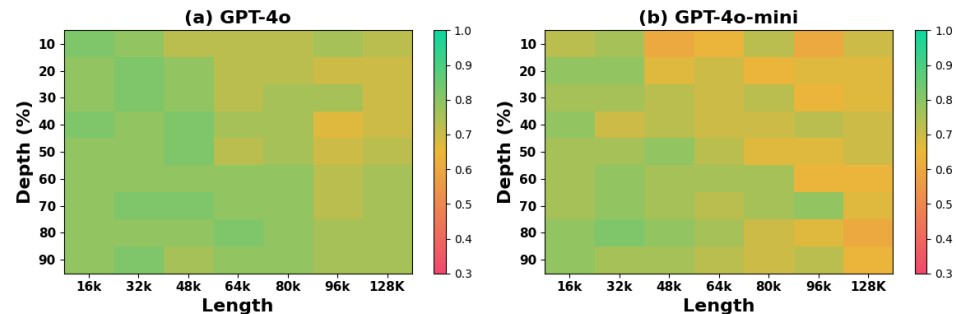

Figure 4: Performance of GPT-4o and GPT-4o-mini on single-document tasks (SSSD, MSSD) with varying placement depths and input lengths. The $y$-axis represents the depth of the relevant document. For example, 10% depth indicates that the document is placed at the first 10% of the input noisy text.

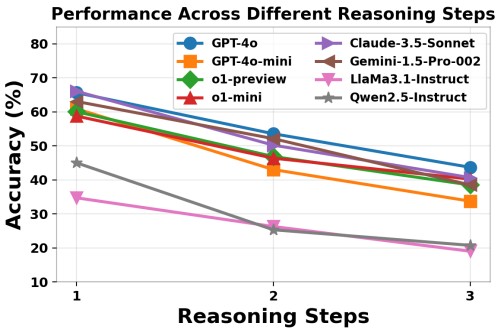

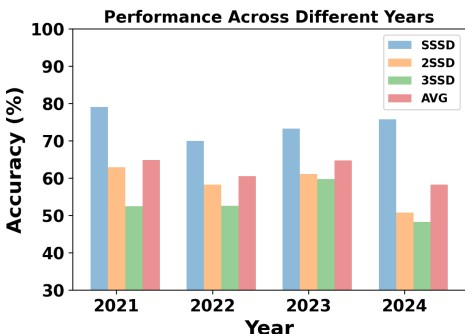

Figure 5: Performance of models at the input length of 32K across varying reasoning steps.

Figure 6: Performance of GPT-4o at the input length of 32K across varying time periods.

as input length increases. This trend suggests that longer inputs introduce more noise, limiting the ability of even advanced LLMs to accurately extract relevant information and reason effectively.

### 4.3 ANALYSIS

**Impact of Placement Depths and Input Lengths.** Figure 4 illustrates the performance of GPT-4o and GPT-4o-mini on single-document tasks with varying document placement depths and input lengths. The results show that smaller placement depths and longer input lengths lead to reduced performance, highlighting the challenge of processing relevant information that is farther from the target question among more noisy context. Notably, GPT-4o-mini demonstrates greater instability with longer input lengths, suggesting that even advanced models may struggle with extreme long inputs. These findings indicate that both insufficient context and excessive noisy text can significantly affect model robustness when handling varying input lengths and document positions.

**Impact of the Number of Reasoning Steps.** Figure 5 illustrates the accuracy of models with an input length of 32k across tasks requiring 1, 2, and 3 reasoning steps. A common trend observed among all models is a decrease in accuracy as the number of reasoning steps increases. GPT-4o demonstrates the highest performance in handling complex multi-step tasks, followed closely by Claude-3.5-Sonnet and Gemini-1.5-Pro-002. In contrast, the other models, particularly LLaMA-3.1-Instruct and Qwen-2.5-Instruct, demonstrate steeper declines in accuracy, suggesting they are less adept at handling tasks that require multiple reasoning steps.

**Impact of Time Period** We aim to assess whether documents collected from queries over different years may impact performance, particularly to explore if more recent documents pose a greater challenge due to potential contamination avoidance. Figure 6 shows the model performance across single-document tasks (SSSD, 2SSD, 3SSD) from 2021 to 2024. While 2SSD and 3SSD display a

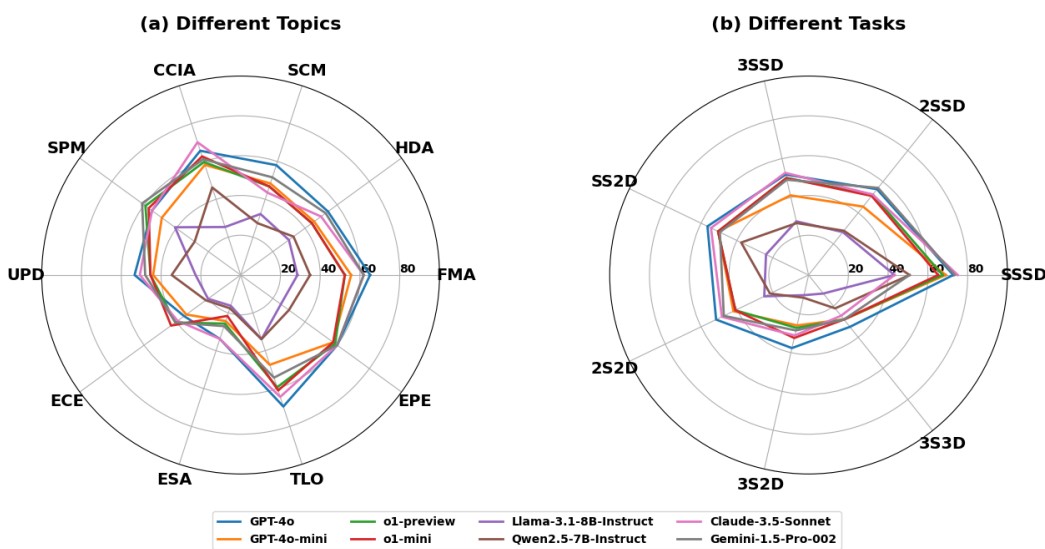

Figure 7: Model performance on the MATHHAY benchmark at 32K across different topics and tasks.

gradual performance decline over time, SSSD remains relatively stable. This indicates that the time period could influence model accuracy, but the effect is uncertain and not clearly confirmed by the current analysis. Further experiments are needed to investigate this hypothesis more thoroughly.

**Analysis of Models across Topics and Tasks.** Figure 7 compares the performance of eight models on the MATHHAY benchmark across (a) different topics and (b) different tasks. In Figure 7(a), GPT-4o generally outperforms the other models across most topics, such as SCM and HDA, with the largest coverage. Claude-3.5-Sonnet and Gemini-1.5-Pro-002 perform similarly but fall behind GPT-4o. LLaMA3.1 and Qwen2.5 perform noticeably lower across all topics. In Figure 7(b), GPT-4o also excels across various tasks, particularly in the SSSD and 3S2D (Three-Step, Two-Document) tasks, maintaining strong accuracy. The smaller models—GPT-4o-mini and o1-mini show similar trends but generally underperform relative to GPT-4o. Again, LLaMA3.1 and Qwen2.5 struggle, especially in the multi-step tasks (e.g., 3SSD, 3S2D, and 2D2D), further indicating their difficulty in handling complex reasoning. Overall, GPT-4o, Claude-3.5-Sonnet, and Gemini-1.5-Pro-002 demonstrate superior robustness across both different topics and tasks, while open-source models show much weaker performance, particularly in complex, multi-step tasks.

## 5 CONCLUSION

In this work, we introduced MATHHAY, a benchmark specifically designed to evaluate the long-context mathematical reasoning abilities of LLMs. MATHHAY is built to challenge LLMs with real-world scenarios requiring both complex reasoning and numerical computation across varying input lengths and document depths. The experimental results show that while Gemini-1.5-Pro-002 performs the best, achieving 51.26% accuracy on tasks with input lengths up to 128K tokens, there remains a substantial performance gap, indicating significant room for improvement. Our findings further reveal that open-source models struggle considerably compared to closed-source counterparts, particularly in tasks that require multi-step reasoning over multiple documents. This underscores the challenges that LLMs face when dealing with noisy and irrelevant information in long contexts, making MATHHAY a crucial benchmark for driving future advances in long-context mathematical reasoning. MATHHAY also offers a novel and automated framework for constructing benchmark datasets, with strong correlations between human-verified and unverified data. This automation enables scalable and efficient testing for future LLMs. MATHHAY aims to drive the development of models with enhanced reasoning capabilities for complex, real-world mathematical tasks.

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

## A PROMPT EXAMPLES

All prompts used in MATHHAY construction consist of two key components: a prompt template and an output parser. The output parser enables users to define any Pydantic model and query LLMs for outputs that adhere to the specified schema.

### A.1 PROMPTS USED IN MATHHAY CONSTRUCTION

#### A.1.1 PROMPT FOR TOPIC GENERATION

---

**Prompt for Topic Generation**

**Prompt Construction:**

```python
from pydantic import BaseModel, Field
from typing import List
from langchain_core.output_parsers import PydanticOutputParser
...
class TopicGeneration(BaseModel):
    topic_list: List[str] = Field(description="A Python list where each element is a string
            representing a single topic. The list should only contain the topics, without any
            additional information or descriptions. Each topic should be concise.")
parser = PydanticOutputParser(pydantic_object=TopicGeneration)
prompt_template.format(format_instructions=parser.get_format_instructions())
```

**Prompt Template:**
You are tasked with generating a diverse set of topics for a benchmark designed to evaluate large language models' abilities in mathematical and numerical reasoning within real-world scenarios.
The goal is to create topics where documents will contain ample numerical data and rich contextual information that can support complex reasoning tasks.
The topics should span various real-world domains where mathematical reasoning is often required, such as: Financial Market Analysis.
For each main topic, ensure that there is potential for generating subtopics that involve mathematical reasoning with substantial numerical content.
Please provide 10 main topics that fit these criteria and briefly describe how each topic can support tasks involving mathematical reasoning and numerical analysis in realistic contexts.

{formatted_instruction}

---

Figure 8: Example prompt for asking the LLM to generate 10 topics.

### A.1.2 PROMPT FOR SUBTOPIC AND QUERY GENERATION

---

**Prompt for Subtopic and Query Generation**

**Prompt Construction:**

```
from pydantic import BaseModel, Field
from typing import List
from langchain_core.output_parsers import PydanticOutputParser
...
class SubtopicAndQueryGeneration(BaseModel):
    subtopic_and_query_map: Dict[str, List[Dict[str, List[str]]]] = Field(
        description="A dictionary where each key is a main topic and its value is a list of
            dictionaries, each containing a 'subtopic' and a list of 'queries'."
    )
parser = PydanticOutputParser(pydantic_object=SubtopicAndQueryGeneration)
prompt_template.format(format_instructions=parser.get_format_instructions())
```

**Prompt Template:**

You are tasked with generating subtopics and corresponding queries for a benchmark designed to evaluate large language models' abilities in mathematical and numerical reasoning within real-world scenarios. Your goal is to create subtopics and queries that are not only relevant but also provide ample opportunities for models to engage in complex numerical analysis and mathematical reasoning.

Instructions:

1. For each main topic provided, generate 4 relevant subtopics.

2. For each subtopic, generate 5 detailed queries, ensuring each query requires reasoning with numerical data extracted from common documents within the specified time period January 2024 to August 2024.

3. Each query should specify both the relevant entities and the time period.

Examples of domains and queries (example time period is May and August 2024):
- Financial Market Analysis:
- Subtopic: Trends in Stock Prices
- Query 1: What was the percentage change in Nvidia's stock price between May 2024 and August 2024?
- Query 2: How did Tesla's stock volatility in April 2024 compare to that in July 2024?

Ensure each query reflects a realistic and complex scenario that necessitates mathematical reasoning to derive the correct answer. The queries should align with the specified time period March 2024 to September 2024 and be formulated to challenge the large language models' numerical reasoning capabilities.

Main topic: Financial Market Analysis

{formatted_instruction}

---

Figure 9: Example prompt for asking the LLM to generate subtopics and corresponding queries.

### A.1.3 PROMPT FOR SSSD QUESTION GENERATION

---

**Prompt for SSSD Question Generation**

**Prompt Construction:**

```
...
class QuantityCell(BaseModel):
    quantity_cell: Tuple[str] = Field(
        description="A tuple containing details about a specific object, including the
            nouns of the object, its attributes, numerical values, relevant dates, and
            locations.")
class ReasoningTask(BaseModel):
    relevant_quantity_cells: List[QuantityCell] = Field(
        description="A collection of QuantityCells.")
    question: str = Field(
        description="A question generated from a subset of QuantityCells. The question
            should involve a single computational step, challenging the model to deduce
            the answer through reasoning.")
    solution: str = Field(
        description="A Python function that solves the generated question using basic
            arithmetic operations. The function must be executable, with clearly named
            variables reflecting the extracted information and a result assigned to a
            variable named `answer`.")
    steps: int = Field(
        description="How many operations(+, -, *, /), i.e., computational steps in python
            solution.")
    answer: float = Field(
        description="The final numerical answer to the question, presented as an Arabic
            numeral. This value is computed by the Python solution and represents the
            correct outcome of the reasoning task.")
class ReasoningTaskList(BaseModel):
    quantity_cells: List[QuantityCell] = Field(
        description="A collection of QuantityCells that represent the extracted numerical
            information, relevant objects, their attributes, and any associated dates or
            locations from the document. This field serves as the basis for generating the
            question and its corresponding solution.")
    tasks: List[ReasoningTask] = Field(
        description="A list of ReasoningTask elements, where each entry contains '
            quantity_cells', 'question', 'solution', and 'answer'. The list should consist
            of at least 3 different ReasoningTask elements.")
parser = PydanticOutputParser(pydantic_object=ReasoningTaskList)
prompt_template.format(document=doc, format_instructions=parser.get_format_instructions())
```

**Prompt Template:**

Your task is to generate a mathematical reasoning question based on the information contained within a single document, identify the relevant numerical information, solve the question using a Python program, and provide the final numerical answer.

Instructions:

1. Extract Quantity Cells: Identify all relevant numerical details from the document, including objects, their attributes, numerical values, and any related dates or locations.

2. Generate a Question: Create a question that involves a single computational step (+,-,*,/) based on a subset of the identified quantity cells. The question should be factual and exclude numerical values from the quantity cells, testing the model's ability to search and reason through the solution based on this data.

3. Provide a Python Solution: Write a Python function that solves the question using basic arithmetic steps. The function should: - Be executable by a Python interpreter. - Avoid using arguments in the function definition; instead, variables must be named and assigned appropriately. - Utilize necessary formulas to perform computations. - Assign the computed result to a variable named 'answer' and ensure the function returns the 'answer' variable.

4. Determine the Final Answer: The final answer should be presented as an Arabic numeral.

Document: {document}

{formatted_instruction}

---

Figure 10: Example prompt for generating Single-Step Single-Document (SSSD) questions using an LLM. Similar prompts are used for tasks like Multi-Step Single-Document (MSSD), Single-Step Multi-Document (SSMD), and Multi-Step Multi-Document (MSMD).

### A.1.4 PROMPT FOR QUALITY CONTROL

---

**Prompt for Generating Python Solution When Given Question and Relevant Documents.**

**Prompt Construction:**

```python
from pydantic import BaseModel, Field
from typing import List
from langchain_core.output_parsers import PydanticOutputParser
...
class ProblemSolving(BaseModel):
    reasoning: str = Field(
        description="solution process."
    )
    python_solution: str = Field(
        description="A Python function that solves the generated question using one or
            several arithmetic operations. The function must be executable, with clearly
            named variables reflecting the extracted information and a result assigned to
            a variable named 'answer'. The solution demonstrates the reasoning process
            leading to the final answer.")
    answer: float = Field(
        description="The final numerical answer to the question, deduced through reasoning.
            ")
parser = PydanticOutputParser(pydantic_object=ProblemSolving)
prompt_template.format(question=question, quantity_cells=quantity_cells, documents=
    documents, format_instructions=parser.get_format_instructions())
```

**Prompt Template:**
You are tasked with solving a mathematical reasoning question using information from the provided documents.
Use the relevant documents and quantity cells to solve the question. Ensure your solution involves single or multiple computational steps based on the relevant data extracted. Focus on arithmetic operations as required by the question.
Instructions: 1. Provide a Python Solution: Write a Python function that solves the question using basic arithmetic or logical steps. The function should:
- Be executable by a Python interpreter.
- Avoid using arguments in the function definition; instead, variables must be named and assigned appropriately based on the given documents and quantity cells.
- Assign the computed result to a variable named 'answer' and ensure the function returns the 'answer' variable.
2. Determine the Final Answer: The final answer should be presented as an Arabic numeral.
Relevant Documents:
{documents}
Relevant Quantity Cells:
{quantity_cells}
Question:
{question}

Output:
- {formatted_instruction}

---

Figure 11: Example prompt for asking the LLM to Python Solution When Given question, relevant quantities, and relevant documents.

## A.2 PROMPT FOR EVALUATION

---

**Prompt for Evaluation.**

**Prompt Construction:**

```
from pydantic import BaseModel, Field
from typing import List
from langchain_core.output_parsers import PydanticOutputParser
...
class LLMVerification(BaseModel):
    reasoning: str = Field(description="Verification process.")
    output: str = Field(description="Yes or No. Yes means the two solutions are equivalent.
        No means the two solutions are different.")
parser = PydanticOutputParser(pydantic_object=LLMVerification)
prompt_template.format(question=question, solutioin1=solution1, solution2=solution2,
    format_instructions=parser.get_format_instructions())
```

**Prompt Template:**
Your task is to determine if the two given solutions are equivalent in terms of reasoning and final answer.
Solution 1:
{solution1}
Solution 2:
{solution2}
Criteria for equivalence:
1. Both solutions should have the same reasoning steps leading to the final answer.
2. The final numerical answers should be identical.
Please analyze the two solutions and state whether they are the same or different. If different, provide a brief explanation of the discrepancies.
Example:
Solution 1:
def solve():
current_value = 45e9 # $45    billion
projected_value = 400e9 # $400 billion
answer = projected_value - current_value
return answer
Answer1: 355000000000.0

Solution 2:
The current value of the AI chip market is projected to be $45 billion, and it is expected to rise to $400 billion by 2027. To find the difference, we subtract the current value from the projected value: $400 billion - $45 billion = $355 billion.
Answer2: 355.0

Output: Yes

{formatted_instruction}

---

Figure 12: Example prompt for asking the LLM to judge if two solutions are the same.

## A.3 PROMPT FOR SOLVING PROBLEMS IN MATHHAY

---

**Prompt for Solving Problems in MATHHAY**

**Prompt Construction:**

```
from pydantic import BaseModel, Field
from typing import List
from langchain_core.output_parsers import PydanticOutputParser
...
class QuantityCell(BaseModel):
    quantity_cell: Tuple[str] = Field(
        description="A tuple containing details about a specific object, including the
            nouns of the object, its attributes, numerical values, relevant dates, and
            locations. This cell encapsulates all information required for extracting and
            computing the answer to the reasoning question."
    )
class ProblemSolving(BaseModel):
    relevant_quantity_cells: List[QuantityCell] = Field(
        description="A collection of QuantityCells that serves as the basis for generating
            the question and its corresponding solution."
    # )
    reasoning: str = Field(description="Solution process.")
    answer: float = Field(description="The final numerical answer to the question, deduced
        through reasoning.")
parser = PydanticOutputParser(pydantic_object=ProblemSolving)
prompt_template.format(question=question, long_context_input=long_context_input, question=
    question, format_instructions=parser.get_format_instructions())
```

**Prompt Template:**

Long-Context Documents:
{long_context_input}

You are tasked with solving a mathematical reasoning question using information from Long-Context Documents. Follow these steps to ensure accurate extraction and calculation:

Instructions:
1. Extract Relevant Numerical Information: Carefully read through the provided Long-Context Documents to identify and list all relevant numerical details. These could include objects, their attributes, numerical values, dates, locations, or any other quantitative data.
2. Analyze and Solve the Question: Use the identified numerical details to solve the given question. Ensure your solution involves a single computational step based on the relevant data extracted. Focus on logical or arithmetic operations as required by the question.

Question:
{question}

{formatted_instruction}

---

Figure 13: Example prompt for asking the LLM to solve the problems in MATHHAY.

## B    TEST DATA EXAMPLES FROM MATHHAY

### B.1    EXAMPLE OF DATA FOR SSSD

---

**Example of Data for SSSD**

**Data Example 1:**
**Topic:** Healthcare Data Analytics
**Subtopic:** Hospital Admission Rates
**Relevant Document:** California Weekly Report Influenza (Flu), RSV, and Other Respiratory Viruses Week 11: March 10, 2024 Ž013 March 16, 2024 Influenza and RSV Highlights 5.0% Influenza positivity 4.0% Outpatient ILI activity 0.2% Hospital flu admissions 570 (+11) Deaths since 10/1/23 (new) 1.6% RSV positivity Influenza Activity Levels+ Geographic Area Activity Level California Statewide Low Northern Region Low Bay Area Region Low Central Region Low Upper Southern Region Low Lower Southern Region Low Key Messages Ŏ0bb Influenza activity is low. Ŏ0bb The majority of detected influenza viruses are A (H1N1)pdm09. Ŏ0bb The flu shot is still the best way to protect yourself against flu, its potentially serious complications, and reduce strain on our healthcare system.
... and 20 deaths among persons with RSV admission diagnoses.
- 295 RSV-coded deaths identified to date for the 2023Ž0132024 season.
Other Respiratory Viruses Surveillance:
- Adenovirus: 5.4% (up from 4.6%)
- Coronavirus (non-SARS-CoV-2): 6.1% (down from 7.2%)
- Enterovirus/Rhinovirus ...
**Question:** What is the total number of deaths from influenza and RSV identified to date for the 2023-2024 season?
**Answer:** 865

---

**Data Example 2:**
**Topic:** Climate Change Impact Assessment
**Subtopic:** Temperature Variations
**Relevant Document:** ...  August 2024 Ž013 Surface air temperature and sea surface temperature highlights:Ŏ0a0
Global TemperaturesŎ0a0
August 2024 was the joint-warmest August globally (together with August 2023), with an average ERA5 surface air temperature of 16.82Ŏ0b0C, 0.71Ŏ0b0C above the 1991-2020 average for August.Ž02fŎ0a0
August 2024 was 1.51Ŏ0b0C above the pre-industrial level and is the 13th month in a 14-month period for which the global-average surface air temperature exceeded 1.5Ŏ0b0C above pre-industrial levels. *Ŏ0a0
The global-average temperature for the past 12 months (September 2023 Ž013 August 2024) is the highest on record for any 12-month period, at 0.76 Ŏ0b0C above the 1991Ž0132020 average and 1.64 Ŏ0b0C above the 1850Ž0131900 pre-industrial average. These values are identical to those recorded for the previous two 12-month periods, ending in June and July 2024...
**Question:** What is the difference between the global-average temperature for the past 12 months above the 1991-2020 average and the global-average temperature for the past 12 months above the pre-industrial average?
**Answer:** 0.88

---

Figure 14: Examples of data for the Single-Step Single-Document (SSSD) task.

## B.2 Example of Data for MSSD

---

**Example of Data for MSSD**

**Data Example 1:**
**Topic:** Financial Market Analysis
**Subtopic:** Trends in Stock Prices
**Relevant Document:** The result is $108,406 million, which is roughly one third of the JPMorgan estimate. I suggest a reason why the JPMorgan estimate of enterprise value could be three times the estimate from the simple formula: The JPMorgan analysts assume that Tesla will earn much more than its cost of capital going forward. Is the assumption reasonable? The evidence presented below suggests not.
Tesla˘2019s Return On Invested Capital
Prior to 2020, Tesla˘2019s return on invested capital was negative. In 2020, it barely turned positive. However, in 2021, it rose to 14%, then to 23% in 2022 and then dropped slightly to 20% in 2023. Therefore, in the last three years, Tesla did indeed earn more than its cost of capital.
However, Tesla˘2019s situation has changed. The JPMorgan analysts gave Tesla˘2019s stock a recommendation of Underweight, indicating that Tesla˘2019s deteriorating fundamentals relate to decreased demand for its vehicles, not decreased supply.
**Question:** What is the average ROIC for Tesla over the years 2021, 2022, and 2023?
**Answer:** 19.0

---

**Data Example 2:**
**Topic:** Climate Change Impact Assessment
**Subtopic:** Temperature Variations
**Relevant Document:** 10 assists as the Warriors (44-35) won for the eighth time in their past nine games. LeBron James scored 33 points and dished out 11 assists and Austin Reaves had 22 points for the Lakers (45-35), who lost consecutive games for just the second time since the start of February. It was Los Angeles˘2019 final home game of the regular season. The Lakers were playing without Anthony Davis, who took a blow to the side of the head in Sunday˘2019s loss to the Minnesota Timberwolves and still was experiencing nausea with a headache on Tuesday. Rui Hachimura supplied 20 points and 11 rebounds and D˘2019Angelo Russell scored 14 points for the Lakers, who had won nine of 10 games before dropping the last two, both at home. The ninth-seeded Lakers are now just a half-game ahead of the No. 10 Warriors. The No. 9 and 10 seeds face off in the play-in tournament, with that winner set to go up against the loser of the 7-8 game for the final Western Conference playoff spot. After an efficient first quarter where they went 7-for-10 from three-point range, the Warriors took a 38-29 lead. After making eight more triples in the second quarter, the Warriors had a 71-60 lead at halftime. The Warriors made 15 of their 22 attempts from three-point range in the first half (68.2%).
**Question:** What is the total number of points scored by LeBron James, Austin Reaves, and Rui Hachimura combined?
**Answer:** 75

Figure 15: Examples of data for the Single-Step Single-Document (MSSD) task.

## B.3 Example of Data for SSMD

---

**Example of Data for SSMD**

**Data Example 1:**

**Topic:** E-commerce Sales Analysis

**Subtopic:** Customer Acquisition and Retention

**Relevant Document 1:** ... Q2 was another strong quarter for eBay as we exceeded expectations across our key financial metrics,¨said Steve Priest, Chief Financial Officer at eBay. ¨We achieved positive year-over-year GMV growth, driven by our execution against strategic initiatives, despite an uneven discretionary demand environment in our major markets.¨

Second Quarter Financial Highlights

Revenue was $2.6 billion, up 1% on an as-reported basis and up 2% on a foreign exchange (FX) neutral basis. Gross Merchandise Volume (GMV) was $18.4 billion, up 1% on an as-reported and FX-Neutral basis. GAAP net income from continuing operations was $226 million, or $0.45 per diluted share. ...

**Relevant Document 2:** ... For example, imagine you started the year with 700 customers but somehow lost 50 by July. Your churn rate is 50/700 X 100 = 7% How to calculate the revenue churn rate? To calculate revenue churn, divide the net revenue lost from existing customers in a given period by the total revenue at the beginning of the period. For example, if your March loss from downgrades is $4,000 while the MRR is $80,000, your revenue churn is 0.05. You can calculate your revenue churn monthly or annually. Having both numbers provides a more nuanced and complete picture of customer retention. It also helps in identifying short-term trends, setting long-term goals, benchmarking performance, and making critical decisions to improve customer retention ...

**Question:** What is the ratio of eBay's Q2 2024 revenue to the monthly revenue in March 2024?

**Answer:** 32500.0

---

**Data Example 2:**

**Topic:** Supply Chain Management

**Subtopic:** Demand Forecasting

**Relevant Document 1:** ... impair the carrying value of the Gillette trade name intangible asset and higher non-core restructuring charges. Core net earnings per share increased by 12% to $6.59. Currency-neutral core EPS increased 16% versus the prior year EPS. The Company generated operating cash flow of $19.8 billion and net earnings of $15.0 billion for the fiscal year. Adjusted free cash flow productivity was 105%, which is calculated as operating cash flow less capital spending and certain other items, as a percentage of net earnings excluding the Gillette impairment charge and ...

**Relevant Document 2:** 136+ The CocaŽ011Cola Company has been refreshing the world and making a difference for over 136 years. Explore our Purpose & Vision, History and ... Comparable EPS (Non-GAAP) Grew 10% to $0.49; Full Year EPS Grew 13% to $2.47; Comparable EPS (Non-GAAP) Grew 8% to $2.69 Cash Flow from Operations Was $11.6 Billion for the Full Year, Up 5%; Full-Year Free Cash Flow (Non-GAAP) Was $9.7 Billion for the Full Year...

**Question:** What is the difference in operating cash flow between P & G for fiscal year 2024 and Coca-Cola for the full year 2023?

**Answer:** 8.2

---

Figure 16: Examples of data for the Single-Step Single-Document (SSMD) task.

## B.4 Example of Data for MSMD

---

**Example of Data for MSMD**

**Data Example 1:**
**Topic:** Sports Performance Metrics
**Subtopic:** Team Performance in Basketball
**Relevant Document 1:** ... April 9 (Wednesday, April 10, Manila time). Golden State made a season-high 26 three-pointers (on 41 attempts), one made triple short of the franchise record, and won the season series with three wins in four games. 26 THREESHere's every single one of 'em Ž6140f pic.twitter.com/VzAsr9Pf67 Curry was 6-for-6 from distance and Draymond Green went 5-for-7 as the Warriors delivered the best three-point shooting percentage ...
**Relevant Document 2:** ... Anthony Davis 1.11 - Karl-Anthony TownsKP in a league of his own pic.twitter.com/I58XI7K17E Jrue Holiday: A- HolidayŽ019s 3-point shooting has exceeded expectations (career-high 44 percent), and he sets the tone defensively every night. He always finds clever ways to contribute, whether he takes 20 shots or five. His maturity, consistency and poise set him apart. One area IŽ019d like to see a bit more is playmaking ...
**Question:** What is the difference between the Golden State Warriors' three-point shooting percentage in the game and Jrue Holiday's three-point shooting percentage?
**Answer:** 19.41463

---

**Data Example 2:**
**Topic:** E-commerce Sales Analysis
**Subtopic:** Product Category Performance
**Relevant Document 1:** ... accounting for 37.6% of the U.S. ecommerce market in 2023. AmazonŽ019s average daily sales revenue is $1.6 billion, contributing to a total revenue of $575 billion in 2023. 56% of consumers start their product searches on Amazon. Most of AmazonŽ019s sales come from independent sellers, with more than 60% of all Amazon sales coming from third-party sellers. 68% of Amazon sellers are third-party (3P) sellers. 58% of Amazon sellers are profitable ...
**Relevant Document 2:** ... Trending beauty products A beautiful physical appearance is a desire by many people, and this is why people spend money on trending makeup products. The total revenue from the beauty industry amounted to $579.20 billion in 2023 and is expected to multiply in the coming years.4 Another report stated that women in the US spend an average of $3,756 annually on beauty products.5 Beauty products comprise items needed for grooming and beautification, including makeup and skincare. Now, you can add some trending beauty products to your store. ...
**Question:** What is the combined total revenue of Amazon and the beauty industry in 2023, and what percentage of this combined total is Amazon's average daily sales revenue?
**Answer:** 50.597816669554675

---

Figure 17: Examples of data for the Single-Step Single-Document (MSMD) task.

