# OpenReview forum: "MathHay: An Automated Benchmark for Long-Context Mathematical Reasoning in LLMs"
_ICLR.cc/2025/Conference — ICLR 2025 Conference Withdrawn Submission_

### Official Review · Reviewer_37d2 · 2024-11-01

**Soundness:** 1
**Presentation:** 3
**Contribution:** 1
**Rating:** 3
**Confidence:** 4

**Summary:**

The paper proposes an LLM-based methodology for generating mathematical benchmarks, composed of a question in some topic which involves a numeric answer which has to be calculated from a text provided as context to the question, together with the expected answer. It also generates a benchmark and evaluates the performance of several LLMs on it.

**Strengths:**

In general, the paper is very well written. In particular, the introduction reads nicely and the goals are clearly and precisely stated.

Looking for an automated methodology for generating benchmarks.

The partition of the difficulty of the tasks with varying number of operations and number of involved documents.

**Weaknesses:**

Related work reads like a survey but it is not clear the actual relation of the contribution of the paper with the cited work. In particular, the contribution with respect to InfiniteBenchAlmost all bibliographic references are from arXiv while there are (newer) published versions of some of the papers.

There is no clear evidence that the actual methodology could be considered to be a contribution. The Architecture shown in Figure 1 is reasonable but it does not seem to be a breakthrough. Several parts of the process are described in very vague way, in particular “Document filtering” which is somewhat critical.

The proposed methodology  does not provision any mechanism for evaluating the quality of the generated questions. The quality control phase only mentions the evaluation of the answers with respect to the questions.

The generated problems does not seem to contain reasoning steps but only the question and the answer.

Experiments focus on evaluating the performance of LLM on the constructed benchmark rather than on evaluating the relevance of the methodology and the benchmark themselves, which are claimed to be the main contribution.

Therefore, there is a lack of adequate experimental evaluation to assess the performance of the methodology to generate relevant benchmarks.

**Questions:**

- Please clarify the differences with InfiniteBench and L-Eval which are the two benchmarks with long contexts and mathematical reasoning.

- What does make your methodology a contribution and the generated benchmark a “unique contribution” to the area?

- Give more details about document filtering, in particular the parameters used (number of distinct numerical values, etc.)

- What do you precisely mean by “high-quality document”? What is the metric used to evaluate it?

- How do you evaluate the relevance of the questions?

- The benchmark is said to be reviewed to ensure the correctness of the reasoning steps, however they are not explicit in the answer. Please explain how this is done.

Other comments

Revise and update references including published versions.

The sentence in Pg. 3, L. 132 “Our benchmark … making it a unique contribution … “ does not appear to be clearly justified.

---

### Official Review · Reviewer_yNkn · 2024-11-04

**Soundness:** 3
**Presentation:** 4
**Contribution:** 3
**Rating:** 5
**Confidence:** 4

**Summary:**

The paper presents a new benchmark for numerical reasoning in long contexts. The benchmark consists of 673 numerical questions, and each question is associated with a set of documents. Some questions can be answered by examining only one document from the set, others need information from multiple documents. Some questions need one arithmetic operation, others need up to 3 steps.

The authors generated the benchmark data semi-automatically, and then evaluated various LLMs with it. They point out that no LLMs were able to answer more than half the questions correctly, leading them to the conclusion that this benchmark would be useful in tracking the capabilities of LLMs.

**Strengths:**

The benchmark and the methodology to generate it are clearly presented.
The experiments show that the benchmark has sufficient value for helping evaluate future LLMs.

**Weaknesses:**

The benchmark is rather small (673 questions), only 20% of which were verified by humans. So in effect, LLMs are evaluating other LLMs!

While the paper repeatedly talks about mathematical reasoning, the benchmark only tests very simple numerical calculations. It does not test true mathematical ability, such as solving problems in a math textbook. Documents are selected if they contain many numbers, as the questions are based on numbers.

The methodology for generating multi-step questions was not described clearly. Since the questions are machine generated, they tend to be rather trivial, and sometimes meaningless, as shown in the few examples in the Appendix.

**Questions:**

Will the benchmark become obsolete quickly? Will you be generating new versions frequently?

Did you use LLMs to filter documents? Some numerical content may be in words, does your filtering step account for that?

When you manually evaluate the answers, what percent of the consistent answers were incorrect? There is a significant gap in the performance on the verified and unverified sets. Also, the key statistics indicate that all the verified questions were single step, so how were the verified stats generated for MSSD and MSMD?

Some typos or unclear statements.
line 51: mathematic reasoning

line 78: within a certain time period to support to form

Page 19 - The documents have strange substrings like 0̆0bb and Tesla2̆019s. Perhaps a copy/paste or LaTeX error?

Line 1065: Figure 15: Examples of data for the Single-Step Single-Document (MSSD) task.

Line 1121: Figure 16: Examples of data for the Single-Step Single-Document (SSMD) task.

Line 1171: Figure 17: Examples of data for the Single-Step Single-Document (MSMD) task.

---

### Official Review · Reviewer_eFAf · 2024-11-04

**Soundness:** 2
**Presentation:** 3
**Contribution:** 2
**Rating:** 5
**Confidence:** 4

**Summary:**

This paper introduces an automated benchmark MATHHAY for accessing the long-context mathematical reasoning capabilities of LLMs. Unlike previous benchmarks like Needle in a Haystack, which focuses primarily on information retrieval within long texts, MATHHAY demands models with both information-seeking and complex mathematical reasoning abilities. Experiments are conducted on 8 top-performing LLMs.

**Strengths:**

* The paper presents a novel approach to evaluating long-context mathematical reasoning in LLMs, addressing a gap in existing benchmarks.
* The paper is well-structured and clearly written. The methodology is explained in detail, and the experimental setup and results are presented in a way that is easy to follow.
* Experiments are conducted on 8 top-performing LLMs. Rich analyses are included.

**Weaknesses:**

Please see the questions below.

**Questions:**

* How does this method ensure the alignment between the question and the Python solution?
* Concern about how to cover if a question has multiple correct solutions, especially based on the quality control strategy?
* Is there an evaluation method to check if the generation question meets the SSSD/MSSD/SSMD/MSMD tasks?
* The evaluation relies on GPT-4o performance, which is not sound enough. Moreover, 100 examples is too small scale for human evaluation.
* In Table 2, the verified two-step questions and three-step questions are 0. Could you please explain this? And what are the standards for selecting the 126 verified questions?

---

### Official Review · Reviewer_XYSn · 2024-11-04

**Soundness:** 2
**Presentation:** 2
**Contribution:** 2
**Rating:** 3
**Confidence:** 3

**Summary:**

The paper proposes a method to generate long-context mathematical reasoning benchmarks for evaluating LLMs' capabilities of information extraction and mathematical reasoning under certain long-context noisy and question-irrelavant circumstance. This paper also evaluates the performance of several LLMs on proposed benchmark MathHay.

**Strengths:**

1. Proposed benchmark in thie paper is tailored  with varying context length, reasoning steps and spectrum noisy or irrelevant content in document to measure  comprehensive performance in LLMs
2. existing method in mathmatical reasoning are difficult to achieve good performance, which reflects the necessity of this benchmark and the high difficulty of mathematical reasoning and information extraction.

**Weaknesses:**

1. Proposed methodology could not be considered as a contribution.The Architecture shown in Figure 1 is not novel while the process is well represented.
2. No quantitative description of the quality of the answers to the questions is given in quality control pipline.
3. The article does not provide specific details of the reasoning steps.
4. The experimental part is not detailed enough, only describing different methods, without showing the connection between the benchmark and existing methods. Therefore, experiment is too simple to be a contribution in this paper.

**Questions:**

1.Will the benchmark be updated in the future?
2. How did you do to filter documents? How  filtering step numerical content  is done?
3. When consistent answers were incorrect?

---

### Official Review · Reviewer_AHoP · 2024-11-04

**Soundness:** 2
**Presentation:** 1
**Contribution:** 1
**Rating:** 3
**Confidence:** 5

**Summary:**

This paper presents MATHHAY, an automated benchmark developed to evaluate the long-context mathematical reasoning capabilities of large language models (LLMs). The benchmark construction involves four main stages: document collection, question generation, quality control, and haystack construction. Real-world mathematical reasoning documents are gathered, and four types of tasks with increasing complexity are generated—ranging from single-step, single-document tasks to multi-step, multi-document tasks. A quality control process ensures data reliability by verifying solution consistency, and a haystack structure is created to challenge models with relevant documents embedded in noisy text. The contributions include: (1) a method to automatically generate a high-quality benchmark for long-context mathematical reasoning, (2) the introduction of MATHHAY with tasks of varying difficulty levels to evaluate reasoning abilities across different input lengths, and (3) experimental findings showing that existing LLMs face significant challenges in handling mathematical reasoning over long contexts, indicating room for improvement in this area.

**Strengths:**

1.	MATHHAY employs an automated quality control and evaluation process, combining exact match and LLM-based assessment, which achieves high correlation with human evaluations. This automation facilitates scalable and efficient testing for future LLM development.
2.	The study provides in-depth insights into how factors like document placement depth, input length, reasoning step count, time period, and topic influence model performance. This multi-angle approach highlights specific challenges for LLMs and provides a nuanced view of model capabilities and limitations.

**Weaknesses:**

1.	The quality control process relies on consistency across multiple solutions generated by the model, but this does not fully ensure mathematical correctness. Consistent solutions may still be incorrect, resulting in potential "false positives" in data quality. Without manual verification or deeper validation, the benchmark may include examples that lack rigor, reducing its reliability for assessing true mathematical reasoning abilities.
2.	This approach relies heavily on adding noise and complex placements to increase task difficulty, which may primarily test the model’s noise filtering rather than its core mathematical reasoning abilities. The setup could overly focus on retrieval challenges, potentially overlooking deeper aspects of mathematical understanding and reasoning.
3.	The benchmark lacks clarity on ensuring that multi-step tasks genuinely require multiple steps. If questions can be solved with either single-step or multi-step solutions, it may lead to inconsistent categorization.
4.	The benchmark does not address how it ensures that generated questions are genuinely related to document content. If the model generates irrelevant or “hallucinated” information, it could compromise the quality and reliability of the generated task.

**Questions:**

1.	Conduct Experiments on Noise Impact: Add experiments that analyze how the model's reasoning ability is affected as the amount of irrelevant (noise) documents gradually increases. This can provide a more nuanced understanding of how noise impacts long-context reasoning performance.
2.	Incorporate Diverse Reasoning Scenarios: Design experiments that assess different types of long-context reasoning skills (e.g., logical deductions, multi-step calculations) across varied noise levels, ensuring that the benchmark evaluates both retrieval and reasoning capabilities in a balanced way.
3.	The paper only introduces four specific combinations for placing three relevant documents in reasoning tasks (First-First-First, Middle-Middle-Middle, Last-Last-Last, and First-Middle-Last). Why doesn’t it explore all possible combinations of placements?

---

### Official Review · Reviewer_Bqb5 · 2024-11-12

**Soundness:** 3
**Presentation:** 3
**Contribution:** 3
**Rating:** 6
**Confidence:** 2

**Summary:**

In this paper, the authors introduce MathHay, a benchmark and data collection pipeline designed to evaluate the mathematical reasoning capabilities of LLMs over extended contexts. Unlike existing benchmarks, MathHay incorporates varied context lengths and introduces multiple documents (ranging from one to three) to assess the models’ ability to filter relevant information from noise. Additionally, it includes multi-step tasks (up to three arithmetic steps) to measure reasoning depth. The benchmark also spans a range of topics, from financial market analysis to agricultural economics. Leading LLMs, including GPT-4, o1-preview, and Gemini-1.5-Pro-002, have been tested, with empirical results indicating that there is still room for improvement in current models.

**Strengths:**

- relatively written paper, easy to follow
- an automated and scalable way to construct a Denmark
- I like the way of using time-sensitive Q&As to mitigate the contamination problem
- the investigation is the impact of time period is interesting, which might motivate further research about the data contamination problem in LLMs.

**Weaknesses:**

- More quantitative and qualitative evaluation may be needed on the quality of the automated generated corpus.

**Questions:**

- I’m curious if the topics and tasks affect the mathematical reasoning performance of LLMs?
- In the classic Needle-in-a-Haystack paper (Kuratov et al., 2023), a pattern emerged where models struggled with retrieval when the ‘needle’ appeared near the beginning but not in the first sentence. However, I haven’t observed any similar pattern in mathematical reasoning—only a general decline in performance as the context window expands, regardless of the needle’s position. This is expected and not particularly insightful. Do you have any educated guesses as to why?”

---

### Note · Authors · 2024-12-02

**Comment:**

Dear Reviewers:

After careful consideration, we have decided to withdraw the manuscript. We sincerely appreciate the time and effort you have devoted to evaluating our work.

Best regards,

Authors

**Withdrawal Confirmation:**

I have read and agree with the venue's withdrawal policy on behalf of myself and my co-authors.